# The Values Encoded in Machine Learning Research

## Abstract

Machine learning (ML) currently exerts an outsized influence on the world, increasingly affecting communities and institutional practices. It is therefore critical that we question vague conceptions of the field as value-neutral or universally beneficial, and investigate what specific values the field is advancing. In this paper, we present a rigorous examination of the values the field advances by quantitatively and qualitatively analysing 100 highly cited ML papers published at premier ML conferences, ICML and NeurIPS. We annotate key features of papers which reveal their values: how they justify their choice of project, which aspects they uplift, their consideration of potential negative consequences, and their institutional affiliations and funding sources. We find that societal needs are typically very loosely connected to the choice of project, if mentioned at all, and that consideration of negative consequences is extremely rare. We identify 67 values that are uplifted in these papers, and, of these, we find that papers most frequently justify and assess themselves based on performance, generalization, efficiency, researcher understanding, novelty, and building on previous work. We present extensive textual evidence and analysis of how these values are concretized. Notably, we find that each of these top values is being defined and applied with assumptions and implications generally supporting the centralization of power. Finally, we find increasingly close ties between these highly cited papers and tech companies and elite universities.

## 1  Introduction

Over the past few decades, ML has risen from a relatively obscure research area to an extremely influential discipline, actively being deployed in myriad applications and contexts around the world. The objectives and values of ML research are influenced by many factors, including the personal preferences of researchers and reviewers, other work in science and engineering, the interests of academic institutions, funding agencies, and companies, and larger institutional and systemic pressures, including systems of oppression impacting who is able to do research. Together these forces influence what research gets done and who benefits from this research. As such, it is important to document and understand the values of the field: what the field is prioritizing and working toward. To this end, we perform a comprehensive analysis of 100 highly cited NeurIPS and ICML papers from four recent years spanning more than a decade.

Our key contributions are as follows:

(1) We develop a fine-grained annotation scheme for the detection of values in research papers, including identifying a list of 67 values uplifted in ML research. To our knowledge, our annotation scheme is the first of its kind, and opens the door to further qualitative and quantitative analyses.

(2) We use our annotation scheme to annotate 100 influential papers and extract their value commitments, which reflect and shape the values of the field more broadly. Like the annotation scheme itself,

the resulting repository of annotated papers is valuable not only in the context of this paper, but also as foundation for further qualitative or quantitative research.[1]

(3) We perform extensive textual analysis to understand some of the dominant values: performance, accuracy, state-of-the-art (SOTA), quantitative results, generalization, efficiency, building on previous work, and novelty (§5). Our analysis indicates that while these values may seem on their face to be purely technical, they are nevertheless socially and politically charged: specifically, we argue that these values are defined and operationalized in ways that centralize power, i.e., disproportionally benefit and empower the already powerful, such as large corporations, while negatively impacting society's least advantaged.

(4) We present a quantitative analysis of the affiliations and funding sources of these most influential papers (§6). We find substantive and increasing presence of big tech corporations. For example, in 2008/09, 24% of these top cited papers had corporate affiliated authors, and in 2018/19 this statistic almost tripled, to 71%. Moreover, of these corporations connected to influential papers, the presence of "big-tech" firms, such as Google and Microsoft, increased more than fivefold, from 11% to 58%.

# 2 Methodology

To understand the values of ML research, we examined the most highly cited papers from NeurIPS and ICML from the years 2008, 2009, 2018, and 2019. We chose to focus on highly cited papers because they reflect and shape the values of the discipline, drawing from NeurIPS and ICML because they the most prestigious of the long-running ML conferences.[2] Acceptance to these conferences is a valuable commodity used to evaluate researchers, and submitted papers are explicitly written so as to win the approval of the community, particularly the reviewers who will be drawn from that community. As such, these papers effectively reveal the values that authors believe are most valued by that community. Citations largely indicate the approval of the community, and help to position these papers as influential exemplars of ML research. To avoid detecting only short-lived trends and enable comparisons over time, we drew papers from two recent years (2018/19) and from ten years earlier (2008/09). We focused on conference papers because they tend to follow a standard format and allow limited space, meaning that researchers must make hard choices about what to emphasize. Collectively, we annotated 100 papers, analyzing over 3,500 sentences drawn from them. In the context of qualitative content analysis, this is a significant effort which allows us to meaningfully comment on the values central to ML.

In more detail, we began by creating an annotation scheme (see below), and then used it to manually annotate each paper, examining the abstract, introduction, discussion, and conclusion: (1) We examined the chain of reasoning by which each paper justified its contributions, which we call the *justificatory chain*, rating the extent to which papers used technical or societal problems to justify or motivate their contributions. (2) We carefully read the text of these sections, individually annotating any and all values from our list that were uplifted or exhibited by each sentence.[3] (3) We documented the extent to which the paper included a discussion of potential negative impacts.

Manual annotation was necessary, both to create the list of values, and to obtain and understand the values present in each paper. Automated approaches, such as keyword searches, would run the risk of systematically skewing the results towards values which are easy to identify, potentially missing or mischaracterizing values which are exhibited in more nuanced ways, or those which were not anticipated. The qualitative approach was key for analyzing the values as well, as it requires a subtle understanding of how the values function in the text and understanding of taken for granted assumptions underlying the values, which methods such as keyword matching would fail to capture.

To assess consistency, 40% of the papers were annotated by two annotators. The intercoder consensus on values in these papers achieved a Cohen kappa coefficient of 61%, which indicates substantial agreement [39]. Furthermore, we used several established strategies to increase consistency, including

---

[1]We include our full set of annotations as supplementary material, along with a CC BY-NC-SA license.

[2]At the time of writing, these two venues, along with ICLR (2013-present), comprised the top 3 conferences according to h5-index (and h5-median) in the AI category on Google Scholar, by a large margin.

[3]We use a conceptualization of "value" that is widespread in philosophy of science in theorizing about values in sciences. In this approach, a value of an entity is a property that is desirable for that kind of entity. For example, speed can be described as valuable in an antelope [28]. Well-know scientific values include accuracy, consistency, scope, simplicity, and fruitfulness [25]. See [27] for a critical discussion of these values.

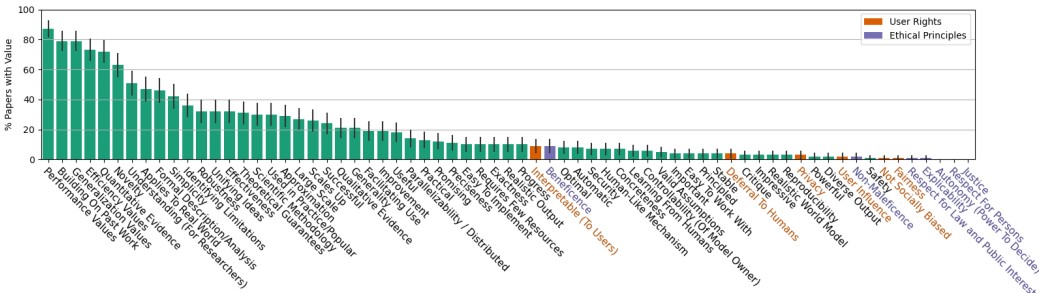

Figure 1: Proportion of annotated papers that uplifted each value.

recoding data coded early in the process [23] and conducting frequent discussions and assessments of the coding process, code list, and annotation scheme [24].

To create the list of values (see Figure 1), we followed best practices in manual content analysis. (1) We began with a list of values we expected to be relevant based on prior knowledge, augmenting this list with seven ethical principles from existing literature [5, 16]. (2) We randomly selected a subset of 10 papers for initial annotation, searching for the values on the list sentence by sentence and adding new values as needed. (3) Through discussion, we revisited all values and produced a final list. (4) We annotated the full set of papers using this list of values, meeting regularly to discuss difficult examples. (5) For the final analysis presented here, we combined some related values into clusters (via consensus), such that they could be discussed together (but are treated separately in the appendix). Formally stated, we establish our codes (short phrases that represent the relevant essence of information, in this case the list of values) using the an inductive-deductive approach. The deductive component involves starting with codes established in existing literature, which ensures we note and can speak to values of interest, including established ethical principles. The inductive component involves the discovery of codes from the data, and impedes inappropriately biased or pre-conceived findings by focusing on emergent codes [7, 24].

The composition of our team also confers validity to our work. We are a diverse team, including undergraduate, graduate, and post-graduate researchers from machine learning, NLP, robotics, cognitive science, and philosophy. This diversity minimizes intra-disciplinary biases, affords the unique combination of expertise required to read the values in ML papers, allows meaningful engagement with relevant work in other fields, and enables best practices including continually clarifying the procedure, ensuring agreement, vetting consistency, reannotating, and discussing themes [24]. Other methods of manual annotation, such as crowd sourcing, lack these advantages.

## 3 Quantitative Summary

In Figure 1, we plot the prevalence of values in 100 annotated papers. The top values are: performance (87% of papers), building on past work (79%), generalization (79%), efficiency (73%), quantitative evidence (72%), and novelty (63%). Values related to user rights and stated in ethical principles appeared very rarely if at all: none of the papers mentioned autonomy, justice, or respect for persons. In Table 1 (top), we show the distribution of justification scores. Most papers only justify how they achieve their internal, technical goal; 71% don't make any mention of societal need or impact, and only 3% make what we considered to be a rigorous attempt at connecting their research to societal needs. In Table 1 (bottom), we show the distribution of negative impact discussion scores. One annotated paper included a discussion of negative impacts and a second mentioned the possibility; none of the remaining 98 papers contained any reference to potential negative impacts. In Figure 3, we show stated ties (funding and affiliations) of paper authors to different institutions. Comparing papers written in 08/09 to those authored 18/19, ties to corporations nearly doubled to 79% of all annotated papers, ties to big tech multiplied over fivefold to 58%, while ties to universities declined to 81%, putting corporations nearly on par with universities in the most cited ML research. In the next sections, we present extensive qualitative examples and analysis of our findings, with additional analyses in the Appendix.

Table 1: Annotation scheme and results for justificatory chain (top) and negative impacts (bottom).

| Justificatory Chain Condition | % of Papers |
| --- | --- |
| Doesn't rigorously justify how it achieves technical goal | 1% |
| Justifies how it achieves technical goal but no mention of societal need | 71% |
| States but does not justify how it connects to a societal need | 16% |
| States and somewhat justifies how it connects to a societal need | 9% |
| States and rigorously justifies how it connects to a a societal need | 3% |

| Negative Impacts Condition | % of Papers |
| --- | --- |
| Doesn't mention negative potential | 98% |
| Mentions but does not discuss negative potential | 1% |
| Discusses negative potential | 1% |
| Deepens our understanding of negative potential | 0% |

## 4 Qualitative Analysis of Justifications and Negative Potential

### 4.1 Justificatory Chain

Papers typically motivate their projects by appealing to the needs of the ML research community, but rarely mention potential societal benefits. Research-driven needs of the ML community include researcher understanding (e.g., understanding the effect of pre-training on performance/robustness, theoretically understanding multi-layer networks) as well as more practical research problems (e.g., improving efficiency of models for large datasets, creating a new benchmark for NLP tasks). Some papers do appeal to needs of the broader society, such as building models with realistic assumptions, catering to more languages, or understanding the world. However, even when societal needs are mentioned as part of the justification of the project, the connection is often loose. Almost no papers explain how their project is meant to promote a social need they identify by giving the kind of rigorous justification that is typically expected of and given for technical contributions.

### 4.2 Negative Potential

Two of the 100 papers discussed potential harms, whereas the remaining 98 did not mention them at all. The lack of discussion of potential harms is especially striking for papers which deal with socially contentious application areas, such as surveillance and misinformation. For example, the annotated corpus includes a paper advancing the identification of people in images, a paper advancing face-swapping, and a paper advancing video synthesis. These papers contained no mention of the well-studied negative potential of facial surveillance, DeepFakes, or misleading videos, respectively.

Furthermore, among the two papers that do mention negative potential, the discussions were mostly abstract and hypothetical, rather than grounded in the negative potential of their specific contributions. For example, authors may acknowledge "possible unwanted social biases" when applying the model to a real-world setting, without discussing the social biases encoded in the authors' proposed model.

## 5 Stated values

The dominant values in ML research, e.g., accuracy or efficiency, may seem purely technical. However, the following analysis of several of these values shows how they can become politically loaded in the process of prioritizing and operationalizing them: sensitivity to the way that they are operationalized, and to the fact that they are uplifted at all, reveals value-laden assumptions that are often taken for granted and may negatively impact the broader society.[4] We thus challenge a conception of prevalent values as politically neural by considering alternatives to their dominant conceptualization that may be equally or more intellectually interesting or more socially beneficial. We have encouraged ourselves, and now encourage the reader, to remember that values once held to be intrinsic, obvious, or definitional have been in many cases transformed over time.

---

[4]Similar points have been made by philosophers of science in the context of the natural and social sciences [25, 27].

Table 2: Random examples of *performance*, the most common emergent value.

| |
|---|
| "Our model significantly outperforms SVM's, and it also outperforms convolutional neural nets when given additional unlabeled data produced by small translations of the training images." |
| "We show in simulations on synthetic examples and on the IEDB MHC-I binding dataset, that our approach outperforms well-known convex methods for multi-task learning, as well as related non-convex methods dedicated to the same problem." |
| "Furthermore, the learning accuracy and performance of our LGP approach will be compared with other important standard methods in Section 4, e.g., LWPR [8], standard GPR [1], sparse online Gaussian process regression (OGP) [5] and $\upsilon$-support vector regression ($\upsilon$-SVR) [11], respectively." |

To provide a sense of what the values we discuss look like in context, we include three randomly selected examples of sentences annotated for each (Tables 2-5), with additional examples in the Appendix. Note that most sentences are annotated with multiple values, but this is not shown here.[5]

## 5.1 Performance

Performance, accuracy, and achieving SOTA form the most common cluster of related values in annotated papers. While it might seem intrinsic for the field to care about performance, it is important to remember that models are not simply "well-performing" or "accurate" in the abstract but always in relation to and as *quantified* by some metric on some dataset. Examining prevalent choices of operationalization reveals political aspects of performance values. First, we find that performance values are consistently and unquestioningly operationalized as correctness averaged across individual predictions, giving equal weight to each instance. However, choosing to use equal weights when averaging is a value-laden move which might deprioritize those underrepresented in the data or world, as well as societal and evaluee needs and preferences. Extensive research in ML fairness and related fields has considered alternatives, but we found no such discussions among the most-cited papers we examined.

Choices of datasets are revealing. They are often driven purely by past work, so as to demonstrate improvement over a previous baseline (see also §5.4). Another common justification for using a certain dataset is applicability to the "real world". Assumptions about how to characterize the real world may also be value-laden. One common assumption is the availability of very large datasets. However, presupposing the availability of large datasets is power centralizing because it encodes favoritism to those with resources to obtain and process them [15]. Further overlooked assumptions include that the real world is binary or discrete, and that datasets come with a predefined ground-truth label for each example, presuming that a true label always exists "out there" independent of those carving it out, defining and labelling it. This contrasts against marginalized scholars' calls for ML models that allow for non-binaries, plural truths, contextual truths, and many ways of being [12, 18, 26].

The prioritization of performance values also requires scrutiny. Valuing these properties is so entrenched in the field that generic success terms, such as "success", "progress", or "improvement" are often used as synonyms for performance and accuracy. However, one might alternatively invoke generic success to mean increasingly safe, consensual, or participatory ML that reckons with impacted communities and the environment. In fact, "performance" itself is a general success term that could have been associated with properties other than accuracy and SOTA.

## 5.2 Generalization

A common way of appraising the merits of one's work in ML is to claim that it generalizes well. Typically, generalization is understood in terms of performance or accuracy: a model generalizes when it achieves good performance on a range of samples, datasets, domains, or applications. Uplifting generalization raises two kinds of questions. First, which datasets, domains, or applications show that the model generalizes well? Typically, a paper shows that a model generalizes by showing that it performs well on multiple tasks or datasets. However, the choice of particular tasks and datasets is

---

[5]To avoid the impression that there is anything unusual or special about these randomly chosen example sentences, we omit attribution, but include a list of all annotated papers in the Appendix.

Table 3: Random examples of *generalization*, the third most common emergent value.

| |
|---|
| "The range of applications that come with generative models are vast, where audio synthesis [55] and semi-supervised classification [38, 31, 44] are examples hereof." |
| "Furthermore, the infinite limit could conceivably make sense in deep learning, since over-parametrization seems to help optimization a lot and doesn't hurt generalization much [Zhang et al., 2017]: deep neural nets with millions of parameters work well even for datasets with 50k training examples." |
| "Combining the optimization and generalization results, we uncover a broad class of learnable functions, including linear functions, two-layer neural networks with polynomial activation $\phi(z) = z^{2l}$ or cosine activation, etc." |

Table 4: Random examples of *efficiency*, the fourth most common emergent value.

| |
|---|
| "Our model allows for controllable yet efficient generation of an entire news article – not just the body, but also the title, news source, publication date, and author list." |
| "We show that Bayesian PMF models can be efficiently trained using Markov chain Monte Carlo methods by applying them to the Netflix dataset, which consists of over 100 million movie ratings." |
| "In particular, our EfficientNet-B7 surpasses the best existing GPipe accuracy (Huang et al., 2018), but using 8.4x fewer parameters and running 6.1x faster on inference." |

rarely justified; the choice of tasks can often seem arbitrary, and authors rarely present evidence that their results will generalize to more realistic settings, or help to directly address societal needs.

Second, uplifting generalization itself reveals substantive assumptions. The prizing of generalization means that there is an incentive to harvest many datasets from a variety of domains, and to treat these as the only datasets that matter for that space of problems. Generalization thus prioritizes distilling every scenario down to a common set of representations or affordances, rather than treating each setting as unique. Critical scholars have advocated for valuing *context*, which stands at the opposite side of striving for generalization [14]. Others have argued that this kind of totalizing lens (in which model developers have unlimited power to determine how the world is represented) leads to *representational* harms, due to applying a single representational framework to everything [13, 1].

Finally, the belief that generalization is even possible implicitly assumes a conservative approach in which new data will be sufficiently similar to previously seen data. When used in the context of ML, the assumption that the future resembles the past is also normative and often problematic as past societal stereotypes and injustice can be encoded in the process [33]. Furthermore, to the extent that predictions are performative [35], especially predictions that are enacted, those ML models which are deployed to the world will contribute to shaping social patterns. Yet, no papers attempt to counteract this quality or acknowledge its presence.

### 5.3 Efficiency

Efficiency is another common value in ML research. Abstractly, saying that a model is efficient typically means saying that the model uses less of some resource, such as time, memory, energy, or number of labeled examples. In practice however, efficiency is commonly referenced to imply scalability: a more efficient inference method allows you to do inference in much larger models or on larger datasets, using the same amount of resources. This is reflected in our value annotations, where 72% of papers mention valuing efficiency, but only 14% of those value requiring *few* resources. In this way, valuing efficiency facilitates and encourages the most powerful actors to scale up their computation to ever higher orders of magnitude, making their models even less accessible to those without resources to use them and decreasing the ability to compete with them. Alternative usages of efficiency could encode accessibility instead of scalability, aiming to create more equitable conditions for ML research.

### 5.4 Novelty and Building on Past Work

Most authors devote space in the introduction to positioning their paper in relation to past work, and describing what is novel. Mentioning past work serves to signal awareness of related publications, to

Table 5: Random examples of *building on past work* and *novelty*, the second and sixth most common emergent values, respectively.

| Building on past work |
| --- |
| "Recent work points towards sample complexity as a possible reason for the small gains in robustness: Schmidt et al. [41] show that in a simple model, learning a classifier with non-trivial adversarially robust accuracy requires substantially more samples than achieving good 'standard' accuracy." |
| "Experiments indicate that our method is much faster than state of the art solvers such as Pegasos, TRON, SVMperf, and a recent primal coordinate descent implementation." |
| "There is a large literature on GP (response surface) optimization." |

| Novelty |
| --- |
| "In this paper, we propose a video-to-video synthesis approach under the generative adversarial learning framework." |
| "Third, we propose a novel method for the listwise approach, which we call ListMLE." |
| "The distinguishing feature of our work is the use of Markov chain Monte Carlo (MCMC) methods for approximate inference in this model." |

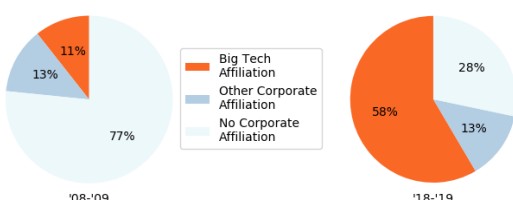

Figure 2: Corporate and Big Tech author affiliations.

establish the new work as relevant to the community, and to provide the basis upon which to make claims about what is new. Novelty is sometimes suggested implicitly (e.g., "we develop" or "we propose"), but frequently it is emphasized explicitly (e.g. "a new algorithm" or "a novel approach").

This combined focus on novelty and building on recent work establishes a continuity of ideas, and might be expected to contribute to the self-correcting nature of science [29]. However, this is not always the case [21] and attention to the ways novelty and building on past work are implemented reveals value commitments. In particular, we find a clear emphasis on technical novelty, rather than critique of past work, or demonstration of measurable progress on societal problems, as has previously been observed [40]. Although introductions sometimes point out limitations of past work (so as to further emphasize the contributions of their own paper), they are rarely explicitly critical of other papers in terms of methods or goals. Indeed, papers uncritically reuse the same datasets for years or decades to benchmark their algorithms, even if those datasets fail to represent more realistic contexts in which their algorithms will be used [6]. Novelty is denied to work that rectifies socially harmful aspects of existing datasets in tandem with strong pressure to benchmark on them and thereby perpetuate their use, enforcing a fundamentally conservative bent to ML research.

## 6 Corporate Affiliations and Funding

Our analysis shows substantive and increasing corporate presence in the most highly-cited papers. In 2008/09, 24% of the top cited papers had corporate affiliated authors, and in 2018/19 this statistic almost tripled, to 71%. Furthermore, we also find a much greater concentration of a few large tech firms, such as Google and Microsoft, with the presence of these "big tech" firms [4] increasing more than fivefold, from 11% to 58% (see Figure 2). The number of most influential papers with corporate ties, by author affiliation or funding, published dramatically increased from 43% in 2008/09 to 79% in 2018/19. In addition, we found paramount domination of elite universities in our analysis as shown in Figure 3. Of the total papers with university affiliations, we found 82% were from elite universities (defined as the top 50 universities by QS World University Rankings, following

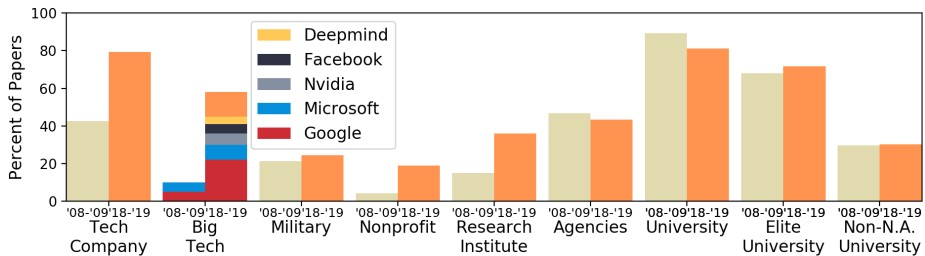

Figure 3: Corporate affiliations and funding ties. Non-N.A. Universities are those outside the U.S. and Canada.

past work [4]). These findings are consistent with previous work indicating a pronounced corporate presence in ML research. In an analysis of 171,394 peer-reviewed papers from 57 major computer science conferences, Ahmed and Wahed [4] show that the share of papers that have at least one corporate affiliated co-author increased from 10% in 2005 for both ICML and NeurIPS to 30% and 35% respectively in 2019. Our analysis shows that corporate presence is even more pronounced in those papers from ICML and NeurIPS that end up receiving the most citations.

The influence of powerful players in ML research is consistent with field-wide value commitments that centralize power. Others have also argued for causal connections. For example, Abdalla and Abdalla [2] argue that the strategies that big tech uses to sway and influence academic and public discourse, closely resemble that of Big Tobacco. Moreover, examining the prevalent values of big tech, critiques have repeatedly pointed out that objectives such as efficiency, scale, and wealth accumulation [33, 34, 19] drive the industry at large, often at the expense of individuals rights, respect for persons, consideration of negative impacts, beneficence, and justice. The top stated values of ML that we presented in this paper such as performance, generalization, and efficiency not only enable and facilitate the realization of big tech's objectives, they also suppress values such as beneficence, justice, and inclusion. A "state-of-the-art" large image dataset, for example, is instrumental for large scale models, further benefiting ML researchers and big tech in possession of huge computing power. A large image dataset that considers negative consequences and is built on the foundations of individual rights and respect for persons, on the other hand, is one that would start with gaining informed consent from the data subject and is considerate of contextual norms over scalability [19]. However, in the current climate where values such as efficiency and scale are a priority, informed consent is perceived as costly and time consuming, evading social needs.

## 7   Discussion

ML research is often perceived as value-neutral, and emphasis is placed on positive applications or potential. This fits into a historical strain of thinking which has tended to frame technology as "neutral", based on the notion that new technologies can be unpredictably applied for both beneficial and harmful purposes [43]. Ironically, this claim of neutrality frequently serves as an insulation from critiques of AI and as a permission to emphasize the benefits of AI [38, 41]. Although it is rare to see anyone explicitly argue in print that ML is neutral, related ideas are part of contemporary conversation, including these canonical claims: long term impacts are too difficult to predict; sociological impacts are outside the expertise or purview of ML researchers [20]; critiques of AI are really misdirected critiques of those deploying AI with bad data ("garbage in, garbage out"), again outside the purview of many AI researchers; and proposals such as broader impact statements represent merely a "bureaucratic constraint" [3]. A recent qualitative analysis of broader impact statements from NeurIPS 2020 similarly observed that these statements leaned towards positive consequences (often mentioning negative consequences only briefly and in some cases not at all), emphasized uncertainty about how a technology might be used, or simply omit any discussion of societal consequences altogether [31].

Importantly, there is a foundational understanding in Science, Technology, and Society Studies (STSS), Critical Theory, and Philosophy of Science that science and technologies are inherently value-laden, and these values are encoded in technological artifacts, many times in contrast to a field's formal research criteria, espoused consequences, or ethics guidelines [44, 10, 8]. There is a long

tradition of exposing and critiquing such values in technology and computer science. Foundationally, Winner [44] introduced several ways technology can encode political values. This work is closely related to Rogaway [37], who notes that cryptography has political and moral dimensions and argues for a cryptography that better addresses societal needs. Weizenbaum [42] argued in 1976 that the computer has from the beginning been a fundamentally conservative force which solidified existing power. In place of fundamental social changes, the computer renders technical solutions that allow existing power hierarchies to remain intact.

Our paper extends these critiques to the field of ML. It is a part of a rich space of interdisciplinary critiques and alternative lenses used to examine the field. Works such as [30, 9] critique AI, ML, and data using a decolonial lens, noting how these technologies replicate colonial power relationships and values, and propose decolonial values and methods. Others [8, 32, 14] examine technology and data science from an anti-racist and intersectional feminist lens, discussing how our infrastructure has largely been built by and for white men; D'Ignazio and Klein [14] present a set of alternative principles and methodologies for an intersectional feminist data science. Similarly, Kalluri [22] denotes that the core values of ML are closely aligned with the values of the most privileged and outlines a vision where ML models are used to shift power from the most to the least powerful. Dotan and Milli [15] argue that the rise of deep learning is value-laden, promoting the centralization of power among other political values. Many researchers, as well as organizations such as Data for Black Lives, the Algorithmic Justice League, Indigenous AI, Black in AI, and Queer in AI, work on continuing to uncover particular ways technology in general and ML in particular can encode and amplify racist, sexist, queerphobic, transphobic, and otherwise marginalizing values [11, 36].

We present this work in part in order to expose the contingency of the present state of the field; it could be otherwise. For individuals, communities, and institutions wading through difficult-to-pin-down values of the field, as well as those striving toward alternative values, it is a useful tool to have a characterization of the way the field is now, for understanding, shaping, dismantling, or transforming what is, and for articulating and bringing about alternative visions.

As with all methods, our chosen approach (careful reading of important sections of highly-cited papers) has limitations. Most notably, this approach does not automatically scale or generalize to other data, which limits our ability to draw strong conclusions about other conferences or different years. Similarly, this approach is less reproducible than fully automated approaches, and for both our final list of values and specific annotation of individual sentences, different researchers might make somewhat different choices. However, given the overwhelming presence of certain values, the high agreement rate among annotators, and the similarity of observations made by our team, we strongly believe other researchers taking a similar approach would reach similar conclusions about what values are most frequently uplifted by the most influential papers in this field. Lastly, we cannot claim to have identified every relevant value in ML. However, by including important ethical values identified by past work, and specifically looking for these, we can confidently assert their relative absence in this set of papers, which we take to be representative of influential work in ML.

## 8 Conclusion and Future Work

We reject the vague conceptualization of the discipline of ML as value-neutral. Instead, we argue that the discipline of ML is inherently value-laden. Our analysis of highly influential papers in the discipline shows that the discipline not only favors the needs of research communities and large firms over broader social needs, but also that it takes this favoritism for granted. The favoritism manifests in the choice of projects, the lack of consideration of potential negative impacts, and the prioritization and operationalization of values such as accuracy, generalization, efficiency, and novelty. All of these overwhelmingly disfavor societal needs, usually without any discussion or acknowledgment. Moreover, we uncover an overwhelming and increasing presence of big tech and elite universities in highly cited papers, which is consistent with a system of power-centralizing value-commitments.

The upshot is that the discipline of ML is not value-neutral. It is socially and politically loaded, valuing and promoting conservative needs at the cost of individuals rights, respect for persons and justice; it increasingly concentrates power in the hands of few already powerful actors; it poses a threat to society's most marginalized by neglecting the potential harms of socially contentions applications of ML.

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
