# OpenReview forum: "The Values Encoded in Machine Learning Research"
_NeurIPS.cc/2021/Conference — NeurIPS 2021 Submitted_

### Official Review · Reviewer_JeHZ · 2021-06-29

**Rating:** 5
**Confidence:** 4

**Summary:**

The paper examines about hundred highly cited machine learning papers published in ICML and NeurIPS conferences  to uncover the "values" embedded in them. In particular, the authors proposed a fine grained annotation scheme for detecting values in ML research, and identify about 67 such values by studying abstract, introduction, discussion, and conclusions sections of the papers under consideration. Using this annotation scheme, they annotated the papers and identified "performance", "accuracy", "state of the art", "quantitative results", "generalization", "novelty" as some dominant values in these papers. The authors also carried out a quantitative analysis of the information pertaining to funding resources and affiliations. The authors argue that all of the values encoded in these papers are defined in a way such that they support power centralization.

**Limitations And Societal Impact:**


Please refer to the previous response for the points concerning limitations.

Some suggestions for improvement:

1. It would be beneficial to include more annotators for every paper to not only make the method more robust, but also to broaden perspectives.

2. Regarding the annotator backgrounds: it might be beneficial to include someone from an HCI discipline also to better inform the process of annotation.

3.  Provide some guidelines or suggest some potential solution pathways towards overcoming some of the issues raised in the paper. For e.g., what proactive steps need to be taken by various stakeholders in order to stop the dominance of the values proposed in the paper?

4. It might be useful to have a section dedicated to related works to distinguish the current paper from prior relevant works.

5. It would be helpful to elaborate on the distinction of the various values, and the process of identifying these as distinct from each other.

**Main Review:**

Strengths:

Significance: Some aspects presented in the paper can be useful in broader settings as listed below.

1. The paper provides an interesting observation concerning some factors that influence and govern ML research. The authors identify about 67 such factors (termed as values in the paper) informed by studies across disciplines. Some of these values can be useful for other research studies.
2. The authors introduce a manual annotation scheme to analyze the motivation for the problem in the papers---the authors refer to this as the justification chain and enlist various possible chains. Similarly, they provide a list of possibilities based on whether negative impacts of the research works have been provided or not. Such annotation schemes can be beneficial checklists.

Clarity:
The paper is presented fairly well written.  There are some minor grammatical issues which can be fixed (e.g. line 55 " they the most...")

Originality:
1. The authors introduce a new (manual) annotation scheme based on the notion of "values". The values themselves ( 67 in number) are identified by means of elaborate discussions between the annotators and analysis of the papers under consideration. The values are identified as entities which can be conceptualized with some desirable properties.

2. A textual analysis of some of the dominant values such as "performance", "building on past research", "novelty", etc. is provided with illustrations from these papers.  Some valuable insights from each of these analyzed values are also provided.


Limitations:

While the paper is both interesting and insightful, there are several aspects which can be significantly improved to address some of the paper's current shortcomings. Some issues of concern are listed below along with some possible suggestions for improvement.

Methodology:

1. The authors argue why a manual annotation is better than an automated analysis, and also mention that the benefit obtained from an automated method was limited.  However, given that at most only two individuals annotated some 40% of the papers, the annotations may be limited in their purview/analysis.  The inter-annotator agreement coefficient provided in the paper is therefore not a reliable indicator of the annotation consistency.

2.  The argument related to choice of papers, years, and conference is somewhat limited. Especially given that the emphasis on ethics, negative consequences, societal impacts and the like have been more pronounced in recent years, it would be beneficial to compare works across years to observe the trends. Even for the years analyzed, it is not clear how the trends shifted, or which years were marked by substantial changes, etc. These aspects are essential for understanding if certain external events and interventions from the community helped or caused harm, and in what way.

3.  The distinction between some of the values enlisted is not clear. For example, what is the difference between "useful", and "practical"? it would be helpful to include examples and definitions or how these values were conceptualized. This also raises the question of how these values were identified and distinguished from each other- more detailed description of the process needs to be provided.

4. For point 4 in the justification chain, how was the aspect "somewhat" defined? In other words, what was the protocol to identify if the paper somewhat justifies its connection to a social need? And in this context, how was it ensured that annotators' bias did not affect the annotation process?

Takeaways:
1. It is important to outline how these observations can inform the ML community in improving things. There have been quite a few works ( some of which the authors mention) that already uncover issues of power imbalances, corporate dominance, funding aspects, and other motivational factors for driving a research. In the light of these, it would be beneficial to provide how the lessons from this study can help in bringing about positive change.  That there is power centralization in ML pipelines is a widely evident and accepted notion, therefore it is not clear what additional lesson can be inferred by the paper in its current form.

Related work:
1. Some relevant literature is not cited, and it is also important to distinguish how the paper is different from some of these other papers. Some of these papers also inform perspectives of graduate students ( like those of the authors in the study) and what roles they can play.

a. Alan Chan, Approaching Ethical Impacts in AI Research: The Role of a Graduate Student

b. Tomo Lacovich, Does Deep Learning Have Politics?

c. Hang Yuan, Claudia Vanea, Federica Lucivero, Nina Hallowell, Training Ethically Responsible AI Researchers: a Case Study

d. Norman Makoto Su and David J. Crandall: The Affective Growth of Computer Vision

Illustrations:
The examples provided in Table 2-5 need some explanations---perhaps, how the particular description highlights a gap and exacerbates a negative value. That way, these tables can be more compelling.

**Time Spent Reviewing:**

5

---

> ### Author Response · Authors · 2021-08-10
> **First Response**
>
> We thank the reviewer for their feedback, and we appreciate their discussion of the paper’s strengths - i.e. its significance, clarity, and originality. We understand that the reviewer has some remaining questions and concerns. Below, we present discussion of the validity of the methodology, the unique contributions of this paper, and the feedback we will incorporate, which we believe addresses all the questions that the reviewer has raised.
>
> Re Methodology Q1: For qualitative textual analysis, including in the view of experts on the team and those that were consulted, 3,500 annotated sentences across 100 papers done by six annotators in multiple fields is viewed as very substantial -- and large enough to be sufficient to demonstrate the broad trends we base our analysis off of. This is further evidenced by the error bars in figure 1 and the inter-annotator agreement. We aren’t sure what is meant by the “inter-annotator agreement coefficient provided in the paper is not a reliable indicator of the annotation consistency”. This is a widely used metric calculated on the jointly annotated documents. We discuss why it is not possible to replace the manual analysis with automated analysis. Manual analysis is required for constructing the list of values that are noted as desirable as well as the in-depth analysis, as discussed in the paper (Sec 2, P3). Many best practices were used and discussed to maximize the completeness of the analysis (Sec 2, P 2-6), and we appreciate and discuss both manual annotation’s strengths and limitations and why it is appropriate for this paper (see Sec 7, P5).
>
> Re Methodology Q2: Because this study is quite original (as noted by R2 and R3), we must lay the foundations for this kind of analysis. Accordingly, we focus on a thorough treatment of the values in highly-cited ML papers at the top ML conferences in the past decade, and we explain why this is a much needed foundational analysis (Sec 2, P1). We share this reviewer’s interest in possible trends across annotators, years and conferences. In our analysis, we found that ethical and societal values were extremely rare even in recent years, among the set of highly-cited papers that we studied. We agree that it would be interesting to see if more striking trends emerge when studying a broader set of papers or conferences including emphasis on the most recent years. The kind of study proposed by the reviewer warrants a thorough analysis of which stated and unstated interventions were adopted by the community at what time, as well as annotation of thousands of additional sentences and hundreds of hours of consensus-based discussions. We agree this constitutes an intriguing direction for future work, and this deserves to be studied in its own right. We kickstart this future work by including in detail our methodology, best practices, annotation scheme, and all annotations (see Sec 2, A.7, and supplementary folder Annotations).
>
> Re Methodology Q3: We appreciate the reviewer’s interest in conceptualization, examples, and clusters of values. We direct the reviewer to Sec 2, in which we discuss our methodology in detail. In particular, see Sec 2, P2 and footnote 3: we use a well-established conceptualization of values. In Sec 2, P5, we present the consensus-based process for identifying and, when necessary clustering values, when needed for qualitative discussion. See the appendix, to see all values treated separately (unclustered). Finally, regarding examples, we direct the reviewer to section A.7, in which we provide 100 random examples, as well as the supplement, in which we make all of our annotations available for inspection by those who are interested in studying this aspect of our study.
>
> Re Methodology Q4: We appreciate the reviewer’s interest in the differences between the justificatory chain codes and the practices taken to minimize bias and maximize completeness. We and the reviewer appear to agree that the first three codes are self-contained/defined. Regarding codes 4 and 5 (papers ‘somewhat’ vs ‘rigorously’ justifying their connection to societal needs), see lines 134-135: ‘Rigorous’ justification constituted “explaining how their project is meant to promote a social need they identify by giving the kind of rigorous justification that is typically expected of and given for technical contributions”. We will move this detail to the section in which we introduce the justificatory chain methodology for reader understanding. Regarding best practices used, see our response to Q1.
>
> Re Takeaways Part 1: We agree with this reviewer that there is clear power centralization in ML pipelines, yet contrary to this reviewer’s statement, this is unfortunately not “a widely evident and accepted notion.” In Section 7 P1, we describe many of the commonly held beliefs that run contrary to this notion of power centralization, and we cite additional references that establish this lack of acceptance. Many ML researchers unfortunately do not have a sense that “Big Tech” is power centralizing, and further much work remains to be done in establishing and understanding the broad and deep mechanisms of this centralization. In fact, much existing work on power centralization has been widely disregarded and even disparaged by influential members and artifacts in the ML community because it does not “speak the language” of ML research papers: compelling papers are often dismissed based on their perceived distance from top ML conferences, their focus on the entire pipeline including applications as opposed to the key artifacts of ML research (papers), and their lack of quantitative results. This paper would be an extremely rare example of overcoming all three hurdles and would provide thorough evidence that the papers we/the reviewer mention can use to establish the significance of their work. This paper additionally seeks to understand how research, and especially highly technical or theoretical research, may contribute to this centralization of power. Authors of theoretical or technical papers commonly state that their work is far removed from societal impacts and thus either has no impacts or the impacts are too difficult to understand. This paper bridges this challenging gap by first analyzing papers for common values or goals, and then connecting these values to societal impacts, providing a valuable tool for understanding the societal impacts of even highly technical or theoretical works. Further, while the reviewer’s comment appears to discuss the impacts of the conclusions of this paper, the contributions of this paper are as much about contributing methodology, data, and evidence (as articulated in our ‘Key Contributions’). We appreciate the reviewer inviting us to highlight our big picture contributions to the field, and we will incorporate this into the paper.
>
> Re Takeaways Part 2: While it is tempting to include recommendations (afterall, we found ‘critique’ to be rarely uplifted/valued in the top papers we studied), after thorough deliberation, we choose to follow the pattern of leading interdisciplinary research in this area by valuing analysis in its own right. (E.g. The importance of analysis as independent from recommendations is well-established in the foundational text Race and Technology). Rather than trying to enforce a particular perspective on goals and solutions, as articulated in lines 318-320, this study is a useful tool to have as a characterization of the way the field is now, for understanding, shaping, dismantling, or transforming what is (not merely improving what is), and for articulating and bringing about alternative visions -- e.g. by uplifting less common values through any number of actions. We believe the importance and nuances of recommendations warrant full-length papers to study this topic in its own right, and broad conversation about what is currently valued by the field and what it would be appropriate to devalue or value, in light of the ways in which ML research is currently impacting society. Although we have many suggestions for encouraging greater emphasis on ethical and societal values, and hope that NeurIPS and ICML choose to do so, there are many stakeholders, and different stakeholders may choose to pursue different paths in which values they foster. Our paper grounds such a conversation in an understanding of the field as it is, and we hope that it will lead to the kinds of recommendations that this reviewer mentions.
>
> Re Related work: We thank the reviewer for these references. We will incorporate them and explain the differences that emerge amongst these papers and our own.
>
> Re HCI: Over the course of this project, we have incorporated team members from many disciplines and have consulted with experts in additional disciplines, including experts in qualitative methods. We have consulted with an HCI expert and incorporated feedback. We will also explicitly name in the paper the fields in which we have consulted experts.
>
> Re additional feedback: Thank you for these fine-grain notes; we will add more detailed explanation of Tables 2-5, and we will clearly delineate the Discussion and Related Work.

---

> > ### Author Response · Authors · 2021-08-28
> > **In addition**
> >
> > After extended discussion, we would like to follow up on two notes --
> >
> > First, we now better understand the recommendation and agree that a “then vs now” comparison (08/09 vs 18/19) is a strong addition, and we commit to including it. More concretely: during our study, we found stratifying Figure 1 and Table 1 by year revealed the years to be remarkably similar, including with regards to societal values like fairness; hence we originally decided to not dedicate space to discussing the stratification. However, as we note above, we certainly shared your interest, and we did indeed examine the stratification during our original analysis; moreover, the stratification is clearly a natural topic of interest, even in the case of finding significant similarity across all years. In sum, following your recommendation, we will include both the updated quantitative plot and table and a qualitative description of the comparison between the two time periods.
> >
> > Second, we understand that there is a tricky duality regarding recommendations; on one hand, our desire to avoid ethics-washing and/or simplistic solutions to complex issues via providing our own list of recommendations, which, on the other hand, might provide helpful first steps to many readers, especially those unfamiliar with the social aspects of machine learning. In our paper, we originally aimed to explicitly state the connection between our detailed mapping of the field as it is and the rich scholarship/advocacy on recommendations (in particular via Sec 7), by referencing that rich literature, which articulates a number of interesting recommendations and possible incentives. After further discussion of the request for recommendations, we believe that there is a best of both worlds: given the reviewer’s approval, we will add a list of  the most relevant recommendations informed by the rich literature we reference.
> >
> > We thank the reviewer for these recommendations.

---

### Official Review · Reviewer_fbBP · 2021-07-14

**Rating:** 7
**Confidence:** 4

**Summary:**

The authors have done a great job of identifying the predominant values in ML research as represented in Neurips and ICML communities. They have identified 67 values and annotated 100 papers over two time periods - 2018/2019 and 2008/2009. They conclude based on their analysis that the dominant values represented are performance, building on past work (in a limited sense of improvement in methodology and performance), generalization, efficiency, quantitative results, and novelty. One of the main arguments the paper makes is that ML is not value-neutral. The authors hope that this will spur conversations in the community about orienting ML towards humanity, and society.

**Limitations And Societal Impact:**

Limitations and societal impact are addressed. Please see above for comments around this.

**Main Review:**

I really enjoyed reading this paper and it was refreshing to see a meta-research article that focuses on important issues that ML research is facing these days. It will be interesting to see this meta-research template adopted and periodical studies like this are conducted every 5 or 10 years.

I appreciate the authors for their painstaking and thorough analysis and careful annotations. Annotating 3500 sentences manually is a significant effort which is appreciated. The methodology adopted here looks fine to me, and the annotations seem to be meticulously done by the authors. My comments are below, and it will be great if the authors clarify:

1. The values picked up by the authors also denote what they think are the most important to analyze - this choice by itself is not neutral, even though it is inspired by previous research in this area. Thoughts on this? During the discussions, did you neglect any potential value for any reason?

2. Even though the annotating team may be diverse in some respects, it may have some shared beliefs that may contribute to some biases. This can be discussed if possible. I do agree that this sort of an intricate exercise cannot be using current crowdsourcing methodologies since it will be impossible to reach meaningful conclusions.

3.  Have you attempted to contextualize the observations in section 3? For example, authors of papers proving abstract theorems may not any clue about their social impact, and it may be far too imaginative to be plausible if they attempt to do so (at least in some cases). However, for papers that develop an image classification method, this may be very much within bounds. There is some mention of this in Sec. 4.2 (paragraph 1), but more details will help.

4. It can also be noted somewhere that the conferences value novelty (and its associated values) more than anything else. It is both top down and bottom up. This is changing slowly though in the recent years. For example, only in recent years conferences have gotten serious about encouraging authors to think about societal impacts of their work. Do you see that papers in recent years have values corresponding to this amplified compared to the older papers? A stratification along the two time periods may help in understanding this.

5. Have you thought about providing some recommendations based on your study? How can the authors of the papers be incentivized to think more about some of the values in the tail of Figure 1? To me it seems like appropriate incentives can move the field in the right direction although it has to be done at the correct pace.

6. Regarding paragraph 1 in Section 5.1: Again some contextualization will help here. ML fairness is a fairly new field and papers in the past decade may not be aware of such ideas. What is more serious is if recent papers that have access to protected group level information do not choose to report fairness measures with respect to those groups.

*Update after end of discussion*

Thanks to the authors for their responses and appreciate their engagement. My score remains the same.

**Time Spent Reviewing:**

4

---

> ### Author Response · Authors · 2021-08-10
> **First Response**
>
> We thank the reviewer for this thorough review. We present responses to their thoughtful questions below and will incorporate our response into the paper!
>
> -Re Q1 and Q2: We appreciate this question, and it is most thoroughly discussed in Sec 2, P2-6. We took several measures to aim to identify all values. For example: we assembled an interdisciplinary team, we used a specific, well-established definition of values, (anything the papers indicated was desirable, footnote 3) rather than properties that our team identified to be  (important or interesting) values; instead of annotating the values in the paper as a whole for a small number of values, which is more likely to have selection bias, we annotated all 3,500 sentences with an unusually high number of codes and double coded 40% of papers; and ​​we collectively discussed any potential values noticed by any member of the team. To the reviewer’s point, we challenged ourselves throughout to annotate even values held to be intrinsic, obvious, or definitional (Sec 5, P1). For example, these practices succeeded in forcing us to treat values like novelty and building on past work (that are often perceived as ‘obvious’ or ‘uninteresting’ even in our socially-informed subfield) as still requiring annotation and discussion. We did not choose to neglect any values we identified. That said, regardless of the number of strategies applied, we agree that no analysis can claim a view from nowhere or simple objectivity (Sec 7, P5), and our team’s most salient shared belief was the a priori belief that analysis of social aspects, values, and priorities of machine learning is important - we will note this explicitly in the paper. If the reviewer sees other beliefs that the author team might share, we are happy to note these as well.
>
> -Re Q3: Indeed, the ML conferences we studied typically contain a mix of theoretical and applied work. In line with the important discussion in response to Q1 (above), we felt it was important to approach this set of papers with no preconceptions about what sorts of considerations we might find in different types of papers, divided along any axes, especially as a central tenet of this paper is that what is considered obvious or impossible for researchers to do is subject to change. There is evidence that even these theoretical papers connect to social impacts: for example, in practice, many theoretical papers include at least some experimental results; the choice to value theory is itself a value-laden choice with social impacts; and there is significant literature indicating that even topics considered at first to be too abstract to contain values (e.g. ‘general-purpose’ technologies or proofs) have ‘politics’ [43] and ‘moral dimensions’ [37]. We have chosen to not buy too heavily into what currently seems impossible, and we see the difficulty of characterizing the values of theoretical work as contributing to the need for research like ours which seeks to challenge this impossibility and understand the societal impacts of even highly technical and theoretical work.
>
> -Re Q4 and Q6: We were also interested in possible trends over time. In our analysis, we found that ethical and societal values including fairness were extremely rare even in recent years, among the set of highly-cited papers that we studied. We agree that it would be interesting to see if more striking trends emerge when studying a broader set of papers including emphasis on the most recent years and stratifying different kinds of papers (e.g. those with certain kinds of datasets, those with protected group level information, etc). We hope that ourselves or others can build upon our work to help answer these questions.
>
> -Re Q5: Since this paper is primarily a quantitative and qualitative study of the current values of the field, we ultimately felt it would decrease the quality of the paper to include our own recommendations. While it is tempting to jump to recommendations (afterall, we found critique to be rarely uplifted/valued in current papers), we follow the lead of leading interdisciplinary research in this subfield (e.g. as summarized thoroughly in Race and Technology): we value analysis in its own right. Rather than trying to enforce a particular perspective, we believe the importance and nuances of recommendations, noted by the reviewer, warrants full-length papers to study this topic in its own right, and a broad conversation about what is currently valued by the field, and what it would be appropriate to devalue or value, in light of the ways in which ML research is currently impacting society. We hope that our paper can contribute to such a conversation, and will lead to the kinds of incentives that the reviewer mentions. We have aimed to support that conversation by including references to a rich literature that we highly recommend, as it articulates a number of interesting recommendations and incentives.
> We appreciate this discussion of several interesting future research directions!

---

> > ### Comment · Reviewer_fbBP · 2021-08-27
> > **thank you**
> >
> > Thanks for your responses. I would still encourage you to discuss:
> > 1. Trends over time, even if it is only qualitative statements.
> > 2. (Im)possibility of providing recommendations and why it may be a larger effort.
> > 3. Challenges in including more annotators in the effort (based on comments from another reviewer).
> >
> > While you may rightfully think that providing recommendations may be too shallow and may seem like "ethics washing", for uninitiated researchers simple recommendations can certainly help move the needle a bit in the right direction. A good example is checklists implemented by Neurips which probably has helped improve reproducibility of work among other things.

---

> > > ### Author Response · Authors · 2021-08-28
> > > **Follow up**
> > >
> > > We thank you for this follow up message and again for further explaining your recommendations. In particular --
> > >
> > > We had originally misread your suggestion as requesting an additional 7 years of annotations to complete a year by year analysis over the course of the intermediary decade, which we felt was beyond the scope but an excellent extension of this paper. We now understand that you were recommending a “then vs now” comparison (08/09 vs 18/19), which upon hearing this we agree is a strong addition, and we commit to including it. More concretely: during our study, we found stratifying Figure 1 and Table 1 by year revealed the years to be remarkably similar, including with regards to societal values like fairness; hence we originally decided to not dedicate space to discussing the stratification. However, as we note above, we certainly shared your interest, and we did indeed examine the stratification during our original analysis; moreover, the stratification is clearly a natural topic of interest, even in the case of finding significant similarity across all years. In sum, following your recommendation, we will include both the updated quantitative plot and table and a qualitative description of the comparison between the two time periods.
> > >
> > > We also thank you for your thoughtful understanding of, on one hand, our desire to avoid  ethics-washing and/or simplistic solutions to complex issues via providing our own list of recommendations, which, on the other hand, might provide helpful first steps to many readers, especially those unfamiliar with the social aspects of machine learning. In our paper, we originally aimed to explicitly state the connection between our detailed mapping of the field as it is and the rich scholarship/advocacy on recommendations (in particular via Sec 7), by referencing that rich literature, which articulates a number of interesting recommendations and possible incentives. After further discussion of the request for recommendations, we believe that there is a best of both worlds: given the reviewer’s approval, we will add a list of the most relevant recommendations informed  by the rich literature we reference. As you describe, for many readers these recommendations will connect our study to strategies for change.

---

### Official Review · Reviewer_CG7U · 2021-07-15

**Rating:** 3
**Confidence:** 3

**Summary:**

The paper investigates ethical values stated in motivations and applications
in machine learning papers. The authors analyzed 100 highly cited ML papers
from four years of two major ML conferences using manual content analysis.
They provide statistics about the frequences of values found in those papers,
the top 5 being performance values, building on past work, generalization
values, efficiency values and quantitative evidence. For four of them they
provide examples and link them to literature that often does not finds them
unproblematic. Also observing that major tech companies currently contribute
heavily to ML research, the authors conclude that these values support
centrailzation of power instead of broader societal wanted agendas.


**Limitations And Societal Impact:**

yes

**Main Review:**

The paper is well written and easy to read. It does not make any methodological
contribution in the area of ML, but it studies ML as an object. It does not provide
any experimental results that could be replicated or falsified beyond some statistics.

The observations, i.e., the values found expressed in the papers as well as the
participation of large tech companies in ML, confirm likely general experience
and statistics, they hardly can be called novel.

The interpretations that link specific techniques or utterances in the literature to
unethical behavior, to me mostly look like non-sequiturs. As an example: "choosing
to use equal weights when averaging is a value-laden move which might deprioritize
those underrepresented in the data or world" (p.5). When ML researchers average
say evaluation measures over instances with equal weights, they certainly do not
"deprioritize" anyone, but they report a statistics about their dataset. The authors
fail to provide evidence that ML researchers claim that their datasets are representative
beyond the context for which they have been collected (say, for the customers of
a company), e.g., for "the world".

I could imagine an interesting panel discussion about these topics, but I do
not see this paper fit as a research contribution on an ML conference.


**Time Spent Reviewing:**

2.5

---

> ### Author Response · Authors · 2021-08-10
> **Initial Response**
>
> We thank the reviewer for their time. However, their evaluation seems to overlook key contributions and points made in the paper and relevant literature.
>
> First, contrary to this reviewer’s claims of non-relevance of this paper topic, our paper topic is significant and relevant to NeurIPS in particular, not only because we intentionally studied papers published in NeurIPS, but also because our paper fits into the larger trend in calls for papers, conference ethics reviews, and protocols, which are pushing toward more social, reflexive, and responsible research. In particular, NeurIPS explicitly invites submissions of interdisciplinary original research on “​​Social Aspects of Machine Learning”. The review does not engage with, and seems to be insufficiently aware of, the evolution of NeurIPS treatment of these topics. Much of the reviewer’s critique seems to be impacted by this lack of familiarity with the subject matter of the paper.
>
> Second, and in particular, the reviewer argues that the paper makes no methodological contributions and does not provide experimental results that could be replicated. However, this assertion does not engage with the discussion regarding this very topic in the paper. As we explain in the introduction, methodology, and appendix: our methodological contribution includes creating a methodology and annotation scheme to detect values in machine learning research papers and can be replicated by others. Our paper provides methodological innovation that is significant in the context of the analysis of social aspects of ML.
>
> Third, the reviewer seems to have misunderstood the goal of the paper. For example, the reviewer claims that “When ML researchers average say evaluation measures over instances with equal weights, they certainly do not ‘deprioritize’ anyone, but they report a statistic about their dataset.” This suggests the reviewer has misunderstood the topic of the paper. As we discuss in the Introduction, paragraph 1, and throughout the paper, this study is not making claims about the personal preferences, priorities, or intentions of individual researchers (see lines 23-28); rather, this paper studies the values and priorities of papers and the field, presenting a rigorous analysis of the ways in which the aspects of models that are uplifted and reported in research papers encode values and priorities regardless of the authors' intentions. Similarly, the reviewer is skeptical about an attempt to “link specific techniques or utterances in the literature to unethical behavior”. However, if by that they mean to be skeptical about ethical judgments of individual machine learning researchers based on some of their research choices they are misunderstanding the paper. Describing the paper as morally evaluating individuals is a mischaracterization of the goal of the paper.

---

> > ### Comment · Reviewer_CG7U · 2021-08-27
> > **Answer to the authors**
> >
> > Thanks for your rebuttal.
> >
> > 1. I agree, I am not a social scientist and not familiar with your research
> >    area. I share the value of contributions from outside the ML community
> >    to the conference, but it implies that once in a while you may get
> >    a reviewer outside your scientific discipline.
> > 2. I talked about "methodological contribution **in the area of ML**", a new
> >    model, a new algorithm, etc. This you do not provide. You may make
> >    a new methodological contribution in the area of social sciences,
> >    which I honestly cannot (and did not) assess.
> > 3. I fear I do not understand what you mean by "values and priorities of papers
> >    and the field". Is it not people who have values? How can a paper have values?
> >    It can express or "encode" values, but whose values are that? The ones
> >    of their authors, right?
> >
> > For me it would be helpful if you could provide some more details about the
> > example I picked from your paper. Authors compute and report averages
> > say of some attribute, say of the age of persons in their dataset.
> > How do they "deprioritize" those not occurring in the dataset? I would
> > assume you would have a hard time finding a useful ML paper stating that
> > only those people in the dataset are of importance, they just say, these
> > are the ones in the very dataset. To me your statements look more like
> > associations, not like thorough implications.

---

### Decision · Program_Chairs · 2021-09-27

**Decision:**

Reject

**Comment:**

The scores of the three reviews are 3, 5, and 7. There is less disagreement than these numbers suggest; all reviewers find the paper worthy but not surprising or deep. As the area chair, I agree. I do have experience with social science research, and this paper lacks insightfulness or originality from that perspective, so I recommend rejection. The paper will eventually be published somewhere, but it won't have great impact.

---

> ### Public Comment · ~Angus_Galloway1 · 2021-11-26
> **Please try to criticize with kindness**
>
> I have no affiliation with this work but I noticed that the meta review only identifies flaws, without mentioning anything positive. Rejection stings enough as is, if you cannot find anything positive to say, how about staying neutral? No details are given about how the authors might address the cons listed; declaring that the work "won't have great impact" is insult to injury. As a past top 10% NeurIPS reviewer, I am also not aware of a requirement that contributions be "surprising or deep", this language does not appear in the Reviewer Guidelines.
>
> I also noticed some of the reviews use language like "The authors fail to...". Language matters, please don't make it personal, make it about the work. Comments like this can have a profound and demoralizing impact, particularly on students.
>
> The approach described here: https://www.themarginalian.org/2014/03/28/daniel-dennett-rapoport-rules-criticism/ has helped me personally to craft better reviews.

---

> > ### Public Comment · Area_Chair_RXXc · 2021-11-27
> > **agreed**
> >
> > Thank you for the feedback, which is valid. In partial response, I would say that the meta-review does call the submission "worthy" and the hundreds of lines of discussion below provide many specific suggestions for the authors. Since NeurIPS is highly selective, reviewers do find most accepted papers to be surprising or deep or unusually good in some other way. And reviewers and meta-reviewers are time-challenged; meta-reviews are the place to explain an overall decision, not to reiterate or even summarize reviews. But thank you again for the reminders.

---

> > > ### Public Comment · ~Subhabrata_Majumdar2 · 2021-11-28
> > > **No place for the last sentence**
> > >
> > > I can sympathize the reviewers and meta-reviewers being time-challenged. All the more reason not to add the rude last sentence that adds zero insight.

---

> > > > ### Public Comment · Area_Chair_RXXc · 2021-11-29
> > > > **an explanation**
> > > >
> > > > As the area chair, let me try to explain the meaning of the last sentence, and what it adds. "The paper will eventually be published somewhere" is part of the reason for the final decision, because if it is true, then readers will be able to read the paper even if it is not published at NeurIPS, so rejecting the paper does not cause harm to potential readers. "It won't have great impact" is also part of the reason for the final decision, because NeurIPS wants to accept high-impact papers, and also because high-impact papers should be published quickly, so that readers can benefit from them.
> > > >
> > > > Area chairs and journal editors have to make decisions. They should explain their decisions so that authors and readers can consider whether or not they agree; they are welcome to disagree with a decision and/or with the reasons for it. Without explanations, authors and readers would be able to learn less from decisions, including learning whether or not they disagree. The last sentence above is part of the explanation for the decision.